# Development of a Bispecific Nanobody Targeting CD20 on B-Cell Lymphoma Cells and CD3 on T Cells

**DOI:** 10.3390/vaccines10081335

**Published:** 2022-08-17

**Authors:** Yanlong Liu, Kexin Ao, Fuxiang Bao, Yi Cheng, Yanxia Hao, Huimin Zhang, Shan Fu, Jiaqi Xu, Qiyao Wu

**Affiliations:** 1College of Veterinary Medicine, Inner Mongolia Agricultural University, Huhhot 010010, China; 2Key Laboratory of Clinical Diagnosis and Treatment Techniques for Animal Disease, Ministry of Agriculture and Rural Affairs (LDTA), Huhhot 010010, China

**Keywords:** CD20, CD3, phage display technology, bispecific single-domain antibody, antitumor effect

## Abstract

B-cell lymphoma is a group of malignant proliferative diseases originating from lymphoid tissue with different clinical manifestations and biological characteristics. It can occur in any part of the body, accounting for more than 80% of all lymphomas. The present study aimed to construct bispecific single-domain antibodies against CD20 and CD3 and to evaluate their function in killing tumor cells in vitro. A Bactrian camel was immunized with a human CD20 extracellular peptide, and the VHH gene was cloned and ligated into a phagemid vector to construct the phage antibody display library. A phage antibody library with a size of 1.2 × 10^8^ was successfully constructed, and the VHH gene insertion rate was 91.7%. Ninety-two individual clones were randomly picked and screened by phage ELISA. Six strains with the high binding ability to human CD20 were named 11, 30, 71, 72, 83, and 92, and induced expression and purification were performed to obtain soluble CD20 single-domain antibodies. The obtained single-domain antibodies could specifically bind to human CD20 polypeptide and cell surface-expressed CD20 molecules in ELISA, Western blot, and cell immunofluorescence assays. The anti-CD20/CD3 bispecific nanobody (BsNb) was successfully constructed by fusing the anti-CD20 VHH gene with the anti-CD3 VHH and the bispecific single-domain antibody was expressed, purified, and validated. Anti-CD20/CD3 BsNb can specifically bind CD20 molecules on the surface of human lymphoma Raji cells and CD3 molecules on the surface of T cells in flow cytometry analysis and effectively mediate peripheral blood mononuclear cells (PBMCs) target Raji cells with a killing efficiency of up to 30.4%, as measured by the lactate dehydrogenase (LDH) method. The release of hIFN-γ from PBMCs during incubation with anti-CD20/CD3 BsNb was significantly higher than that of the control group (*p* < 0.01). The anti-CD20/CD3 BsNb could maintain 80% binding activity after incubation with human serum at 37 °C for 48 h. These results indicated the strong antitumor effect of the constructed anti-CD20/CD3 BsNb and laid the foundation for the further development of antitumor agents and the clinical application of anti-CD20/CD3 BsNb.

## 1. Introduction

B-cell lymphoma is a heterogeneous lymphoproliferative disease derived from lymphoid tissue with different clinical manifestations [1]. It ranges from slow-growing Hodgkin’s lymphoma (HD) to the more aggressive non-Hodgkin’s lymphoma (NHL), the incidence of which has been on the rise in recent years, and B-cell disease accounts for more than 85% of all cases of non-Hodgkin’s lymphoma (NHL) [2,3]. At the same time, studies have shown that more than 95% of B-cell lymphomas exhibit abnormally high expression of CD20 [4]. The CD20 molecule, also known as B lymphocyte surface antigen B1, transmembrane domain 4 subfamily A member 1 and MS4A1, is a specific membrane protein on the surface of B lymphocytes and exists in the nonglycosylated form [5]. This antigen is expressed in pre-B lymphocytes and mature B lymphocytes but not in hematopoietic stem cells, plasma cells, or other tissues [6]. CD20 is composed of 297 amino acids (aa), and its relative molecular weight is 33–35 kDa. The CD20 molecule has four transmembrane domains containing two extracellular rings and intracellular regions. Studies have demonstrated that the 47 aa outer loop between the third and fourth transmembrane regions is an epitope of CD20, which is recognized by most anti-CD20 antibodies [7,8]. The CD20 protein has no known natural ligand to interfere with its binding to antibodies, and there is no significant internalization and shedding of CD20 after binding to antibodies or antigenic modulation due to binding to antibodies, making it an ideal target for the treatment of B-cell lymphoma [9].

According to the expression characteristics of CD20 in the B lymphocyte development stage, it has been selected as one of the targets for the treatment of B-cell lymphoma, and many antibody drugs have been successfully developed based on this target [10]. CD20-targeted monoclonal antibody drugs can be divided into three generations: the first generation of rituximab is a human mouse chimeric monoclonal antibody; the second generation of Ofatumumab is a humanized monoclonal antibody, and the third generation of trastuzumab is a monoclonal antibody with a modified Fc segment. Rituximab is the world’s first monoclonal antibody approved by the FDA for non-Hodgkin’s lymphoma [11]. Monoclonal antibody drugs have greatly improved the treatment status of non-Hodgkin lymphoma. However, they have also led to many adverse reactions and serious side effects. Therefore, studies on reducing the immunogenicity of CD20 antibodies and improving its affinity to Fc receptors were intensified [12]. Roche’s CD3/CD20 bispecific antibody RO7082859 was approved in China on 9 September 2020 for the treatment of diffuse large B-cell lymphoma, further improving the treatment of non-Hodgkin lymphoma.

Heavy chain-only antibodies (HCAbs), a unique dimer antibody type, have been detected in the serum of Camelidae and cartilaginous fish. These antibodies are devoid of light chains, and the CH1 region of the heavy chain is missing so that the antigen is recognized by a single variable domain [13]. This variable domain has been referred to as VHH or, when expressed recombinantly, as a single-domain antibody (sdAb) or nanobody. Its molecular mass of 12–15 kDa is significantly lower than that of conventional antibodies (150 kDa) [14]. A nanobody has three antigen-binding loops known as H1, H2, and H3. The H3 loop is longer than the corresponding loop of human VHs and forms a rigid structure often supported by an interloop disulfide bond. Compared to the concave or flat topology of the paratope of conventional antibodies, the VHH sdAb has a flat or convex paratope to interact with concave-shaped epitopes [15]. SdAbs can be used to display the hidden epitopes of traditional immunoglobulins, making them widely used in immune research, diagnostics, medical and biological imaging, and therapeutic antibody development [16]. The low molecular weight of single-domain antibodies provides better solubility and stability. Compared with conventional antibodies, single-domain antibodies are less immunogenic and easier to humanize and apply in the clinic [17].

Since 1960, when Nisonoff et al. [18] first synthesized bispecific antibodies, numerous additional bispecific antibodies have been developed. Most routinely combine monovalent antibodies to two different targets in the form of IgG. The scheme of these antibodies is designed in such a way that if two different light chains and two different heavy chains are coexpressed in the same cell, up to nine mismatches may be formed between the light and heavy chains, generating a large number of nonfunctional antibodies; thus, the purification of bispecific antibodies of the target form is difficult, and the yield is low [19]. Nanobodies with hydrophilic surfaces do not bind to light chains, which are prone to polymerization, avoid the mismatches between the heavy and light chains of conventional antibodies [20,21], and are characterized by small molecular weight, high solubility, and stability. Bispecific antibodies can be constructed from two nanobodies targeting different targets of tumors by genetic engineering to improve the specificity of antitumor antibodies and make them ideal antibodies.

Bispecific nanobodies (BsNb) targeting tumor antigens and T lymphocytes have been developed as a promising antitumor therapy [22,23]. The BsNb can recognize two different antigens or epitopes at the same time. Compared with nanobodies, a BsNb offers stronger specificity, targeting, and lower off-target toxicity and shows the advantages of double-target signal blocking, reducing immune escape and drug resistance in treatment. The enhancement of BsNb binding affinity, prolongation of serum half-life, and reduction in drug resistance and serious adverse reactions make it a research hotspot in the diagnosis and treatment of infection, tumor, and immunity [23,24].

In this study, a Bactrian camel was immunized with CD20 extracellular peptide to construct a phage display antibody library. Human CD20 extracellular peptide was used as an antigen to enrich and screen specific single-domain antibodies (sdAbs). An anti-CD20-specific single-domain antibody was expressed and purified to determine the specificity of the antibody. Based on the obtained anti-CD20 nanobodies, we further used a genetic engineering method to construct the bispecific nanobody anti-CD20/CD3 BsNb by fusing the anti-CD20 VHH gene with a well-verified anti-human CD3 VHH, and the binding activity and tumor cell killing effect of the anti-CD20/CD3 BsNb were identified by an in vitro test. The study lays the foundation for its application in targeted antitumor research and provides a safer and more effective treatment method for B-cell lymphoma.

## 2. Materials and Methods

### 2.1. Construction of the CD20-Specific Phage Antibody Library

Three-year-old healthy female Bactrian camels were used for human CD20 polypeptide immunization. Bactrian camels were raised in mountainous grassland in the southern suburbs of Hohhot city, Inner Mongolia Autonomous Region, China, and provided free water and food. Five milliliters of blood was collected from the jugular vein before immunization, and the serum was collected as the basic serum after it coagulated naturally and stored at −20 °C. One milligram of polypeptide (KISHFLKMESLNFIRAHTPYINIYNCEPANPSEKNSPSTQYCYSIQS) comprising 47 aa outer loops in the third and fourth transmembrane regions of human CD20 (Sangon Biotech (Shanghai) Co., Ltd., Shanghai, China) was mixed with 1 mL of complete Freund’s adjuvant at a 1:1 ratio and shaken to induce full emulsification. Incomplete Freund’s adjuvant was used for booster immunization, and immunization was conducted once every two weeks. One week after each immunization, blood was collected from the jugular vein and placed at 4 °C to separate the serum for ELISA detection of the antibody titer. Two weeks after the fourth immunization, 50 mL of blood was collected from the jugular vein of the immunized camel, and peripheral blood lymphocytes were isolated. All animal experiments were carried out under the supervision and guidance of the Experimental Animal Use and Care Committee of Inner Mongolia Agricultural University.

After the fourth immunization, 100 mL of blood was collected from the jugular vein of a Bactrian camel, and PBMCs were isolated using the Bovine Blood Lymphocyte Isolation Kit (TBD Science, Tianjin Haoyang Biological Manufacture Co., Ltd., Tianjin, China). Total RNA of PBMCs was extracted with TRIzol reagent (Ambion, Austin, Texas, USA) and reverse transcribed into cDNA using a reverse transcription kit (Promega, Madison, Wisconsin, USA) with random hexamers as primers. The VHH fragment was amplified by nested PCR using the primers listed in Table 1. The first round of PCR amplification used cDNA as a template and P1/P2 as primers to amplify the gene sequence of the leading signal sequence of the CH2 region, which contained the 900 bp (VH-CH1-CH2) and 600 bp (VHH-CH2) fragments. The 600 bp fragment was purified by 1% agarose gel electrophoresis and used as a template for the second round of PCR. The full-length VHH genes from FR1 to FR4 were obtained by the second round of PCR amplification with P3/P4 as primers. The VHH gene fragment was double cleaved by restriction enzymes *Nco* I and *Not* I (New England Biolabs, Hitchin, Hertfordshire, U.K.) and ligated into the same enzyme-digested pMECS (gift of Professor Serge Muyldermans of Vrije Universiteit Brussel, Brussel, Belgium) vector with T4 ligase (New England Biolabs, Hitchin, Hertfordshire, U.K.), and the ligated product was transformed into *E. coli* TG1 competent cells (GE Healthcare, Chicago, Illinois, USA) to construct the phage antibody library against the human CD20 extracellular polypeptide. Calculating by serial dilutions of the transformation mixture and counting the colonies after plating on selective ampicillin-containing plates, twenty-four colonies were randomly selected to monitor the percentage of clones with the appropriate insert size. PCR amplification was performed using the sequencing primers MP57 and GIII (Table 1) to characterize the transformation efficiency of the phage library.

### 2.2. Enrichment and Screening of a Human CD20-Specific Single-Domain Antibody

The phage antibody library was inoculated into a 100 mL 2 × YT medium, and the bacterial solution with an OD600 nm value of no more than 0.3 was placed in a constant temperature incubator at 37 °C and fully shaken at 250 r/min for 2 h. M13K07 helper phage (GE Healthcare, Chicago, IL, USA) was added to the bacteria with a ratio of phage: bacteria = 20:1 for infection. The solution was centrifuged, the supernatant was discarded, 40 mL of 2 × YT-AK liquid medium was used to resuspend the precipitate, and the bacteria were cultured overnight with shaking at 220 r/min at 37 °C. The bacterial solution was placed in a centrifuge tube at 4 °C and centrifuged at 7197× *g* for 25 min. The supernatant was separated, and 16 mL of 20% PEG8000/NaCl was added to the recombinant phage for mixing for 30 min. Centrifugation was performed at 7197 g at 4 °C for 25 min, the supernatant was discarded, and the precipitated phage was resuspended with 1 mL PBS. The recombinant phage was added to a Stripwell™ microplate (Corning, NY, USA) precoated with 10 μg/mL human CD20 extracellular polypeptide and incubated at 37 °C for 1 h. One hundred microliters of 100 mM triethylamine eluting buffer was added to each well and incubated at room temperature for 10 min. Then, 50 μL of 1 M Tris-HCl was added to each well. The eluted recombinant phage was reinfected with *E. coli* TG1, and the helper phage was added for rescue. The same enrichment and screening steps were repeated three times.

The recombinant phage from the third screening was used to infect *E. coli* TG1, which was coated on a 2 × YT-AG solid medium at 37 °C and incubated overnight. The next day, 92 clones were randomly picked to prepare the recombinant phage, and 100 μL PBS was used to resuspend the phages for phage ELISA. The recombinant phage was added to a Stripwell™ microplate precoated with 2.5 μg/mL human CD20 extracellular polypeptide for 2 h at room temperature. The helper phage M13K07 was used as a negative control, and PBS was used as the blank control. One hundred microliters of HRP-labeled anti-M13 monoclonal antibody (GE Healthcare, Chicago, IL, USA) was added to each well and incubated at room temperature for 1 h, and 100 μL of TMB solution (Promega, Madison, WI, USA) was added to each well for 10 min, and the plate was read at OD450 nm in a microplate reader. The absorbance of the experimental group/negative control group ≥2.1 was considered positive.

### 2.3. Expression and Purification of a CD20-Specific Single-Domain Antibody

Fifty microliters of phage ELISA-positive bacterial clone was added to a 5 mL 2 × YT liquid medium overnight culture shaking at 250 r/min at 37 °C. The plasmid was extracted from the bacteria, and the gene sequences of VHH were obtained by sequencing. The VHH fragments were obtained by digesting the plasmids with *Nco* I and *Not* I restriction enzymes and ligated into the pET-25b (+) –SBP plasmid (The pET-25b (+) vector carried a streptavidin binding protein that was constructed and preserved by the Public Health Department of the College of Veterinary Medicine, Inner Mongolia Agricultural University). The products were transformed into *E. coli* BL21 (DE3) competent cells (TransGen Biotech, Beijing, China). The recombinant sdAb antibodies were induced by 1 mM isopropyl-β-D-thiogalactoside (IPTG) (Solarbio Life Sciences, Beijing, China), and then the proteins were purified by the Ni-NTA Sefinose™ Resin (Sangon Biotech (Shanghai) Co., Ltd., Shanghai, China) under natural conditions. The purified recombinant single-domain antibodies were analyzed using SDS–PAGE electrophoresis.

### 2.4. Binding Activity and Specificity of the sdAb

Different concentrations of recombinant sdAb were added as primary antibodies to 96-well Stripwell™ microplates precoated with 2.5 μg/mL human CD20 extracellular polypeptide. The cell lysates of pET-25b-SBP empty plasmid vector-transformed *E. coli* were used as a negative control, and PBS was used as a blank control. The results were determined by using an HRP-conjugated 6 × His-tagged mouse monoclonal antibody (Proteintech Group, Wuhan, China) as a secondary antibody at a concentration of 1:10,000. After adding TMB substrates, the plate was read at OD450 nm in a microplate reader. Meanwhile, a 96-well Stripwell™ microplate was precoated with 2.5 μg/mL Aβ_1-42_ oligopeptide (Sangon Biotech (Shanghai) Co., Ltd., Shanghai, China) in parallel, and the purified sdAbs were added as primary antibodies to perform ELISA to verify the binding specificity of the antibody.

Human CD20 extracellular polypeptides were separated by SDS–PAGE electrophoresis, and the proteins were transferred to polyvinylidene fluoride (PVDF) membranes after electrophoresis. The PVDF membrane was placed in 3% BSA and incubated overnight at 4 °C. The purified sdAb was used as the primary antibody, and HRP-conjugated 6 × His-tagged mouse monoclonal antibody was added as the secondary antibody at a concentration of 1:10,000. The membranes were ultimately visualized using an ECL solution (Solarbio Life Sciences, Beijing, China).

### 2.5. Immunofluorescence Assay

Coverslips were soaked in anhydrous ethanol for 5 min, then clamped and allowed to dry naturally before being placed in a 6-well cell culture plate. The 6-well plate was treated with 0.01 mg/mL polylysine and dried on an ultraclean table. The test group of human B lymphocyte cell line Raji (Newgainbio, Wuxi, China) and the control group HEK293A cells (kindly provided by Professor Jinsheng He of Beijing Jiaotong University) were added to 6-well cell culture plates, respectively, and incubated for 2 h to allow the cells to grow on the glass coverslips. The plates were rinsed 3 times with PBS for 3 min each time. Then, 4% paraformaldehyde (Solarbio Life Sciences, Beijing, China) was added to the plate to fix the cells, and after 20 min at room temperature, each plate was rinsed 3 times with PBS for 3 min each time. Then, 5% BSA was added to the plate at room temperature for 1 h. The purified sdAb was added to the experimental wells as a primary antibody, and PBS was added to the control wells and incubated overnight at 4 °C. The CoraLite^®^488-conjugated 6 × His His-Tag Mouse Monoclonal antibody (Proteintech Group, Wuhan, China) was used as the secondary antibody at a concentration of 1:150 and added to each well for 1 h at room temperature, and the wells were rinsed 5 times for 3 min each in PBST. The samples were placed under a confocal microscope (ZEISS, LSM-800, Oberkochen, Germany) to detect the fluorescence signals.

### 2.6. Construction, Expression, and Purification of Bispecific Nanobodies Targeting CD3 and CD20

The VHH gene fragment of the anti-CD20 single-domain antibody identified in this study was linked to a VHH gene of a human CD3-specific single-domain antibody, which has been well characterized and validated in invention patent WO 2010037838 A2 (Kufer and Raum, 2011, Moradi-Kalbolandi et al., 2019) by a (G_4_S)_3_ linker peptide (linker) using SnapGene Viewer software (GSL Biotech LLC, San Diego, CA, USA) to construct a CD20 sdAb-linker-CD3 sdAb sequence. The constructed gene sequences were sent to Sangon Biotech (Shanghai) Co., Ltd., Shanghai, China, for gene synthesis. The synthesized gene was ligated into pET-22b (+) to obtain the recombinant plasmid pET-22b (+)/BsNb. The recombinant plasmid was transformed into *E. coli* BL21 (DE3) competent cells, and expression was induced by a final concentration of 1 mM IPTG for 6 h. The recombinant antibodies were purified by Ni-NTA Sefinose™ Resin. The expression and purification of recombinant BsNb were analyzed with SDS–PAGE electrophoresis.

### 2.7. Identification of the Binding Activity of Anti-CD20/CD3 BsNb to the CD20 Peptide

Purified anti-CD20/CD3 BsNb was added as a primary antibody in a dilution gradient from 30 μg/mL to 5 μg/mL into a 96-well Stripwell™ microtiter plate precoated with 2.5 μg/mL human CD20 extracellular peptide. pET-22b (+) empty plasmid vector-transformed *E. coli* lysate solution was used as a negative control, and PBS was used as a blank control. The results were determined by using HRP-conjugated 6 × His-tagged mouse monoclonal antibody as a secondary antibody at a concentration of 1:10,000 and adding TMB while avoiding light for color development and reading at OD450 nm by a microplate reader.

### 2.8. Flow Cytometry Analysis

The labeling of antibodies with fluorescein isothiocyanate (FITC) (Sangon Biotech (Shanghai) Co., Ltd., Shanghai, China) was performed according to the manufacturer’s instructions. Briefly, FITC was dissolved in dimethyl sulfoxide (DMSO), added to 1.2 mg/mL anti-CD20/CD3 BsNb, and incubated on a horizontal shaker protected from light for 1 h. The desalting column was precentrifuged to desalinate the coupling product, placed on the resin surface in the desalting column, and centrifuged at 3000 r/min for 2 min. FITC-labeled anti-CD20/CD3 BsNb was stored at 4 °C.

Peripheral blood from a healthy donor diluted with an equal volume of saline was slowly added to a human peripheral lymphocyte isolation kit (TBD science, Tianjin Haoyang Biological Manufacture Co., Ltd., Tianjin, China), followed by centrifugation at 800 g for 30 min. The middle white cloudy mononuclear cells were aspirated into a new 15 mL centrifuge tube and mixed with PBS, centrifuged at 1500 r/min for 10 min, and washed twice with PBS. RPMI 1640 medium (Thermo Fisher Scientific, Waltham, MA, USA) containing 10% fetal bovine serum (FBS) (Thermo Fisher Scientific, Waltham, MA, USA) was used to resuspend the cells and inoculate them into T25 cell culture flasks.

Raji cells and PBMCs were washed twice with PBS containing 1% BSA and divided into EP tubes. RPMI 1640 medium was used to adjust the total number of cells per tube to 1 × 10^6^. In the experimental group, FITC-anti-CD20/CD3 BsNb was added to the Raji cell group, PBMC group, and Raji cell and PBMC mixed group to a final concentration of 100 μg/mL, and 100 μL PBS was added to the control group. The cells were incubated in a rotating ice bath at 4 °C for 60 min and then washed with PBS containing 1% BSA 3 times. Finally, the cells were resuspended in 200 μL PBS and detected on a FACSAria III (BD Bioscience, San Jose, CA, USA), and the data were analyzed using FlowJo (FlowJo Software, Ashland, OR, USA).

### 2.9. LDH Release Assay Detects Anti-CD20/CD3 BsNb-Mediated Cytotoxicity of PBMCs In Vitro

The cytotoxicity of PBMCs mediated by anti-CD20/CD3 BsNb in vitro was determined using an LDH kit (Elabscience, Wuhan, China). The target cells were Raji cells, and the effector cells were PBMCs. The concentration of anti-CD20/CD3 BsNb ranged from 0 μg/mL to 40 μg/mL. Raji cells were adjusted to 1 × 10^4^ cells per well in RPMI 1640 medium and added to a PerkinElmer Cell Carrier, with three parallel wells in each group, and cultured for 12 h. PBMCs and anti-CD20/CD3 BsNb (40 μg/mL, 20 μg/mL, 10 μg/mL, 5 μg/mL, 2 μg/mL, 1 μg/mL, and 0 μg/mL) were added to Raji cells at a ratio of effector: target = 5:1 and effector: target = 10:1, and 100 μL RPMI 1640 medium was added to blank wells. Then, 50 μL of Raji cells + 50 μL of PBMCs were added to sample control wells, and 50 μL of Raji cells + 50 μL of PBMCs + 10 μL of lysate was added to maximum enzyme activity control wells and incubated for 24 h at 37 °C. Then, 10 μL of lysate was added to the target cell maximum release wells 1 h before the end of incubation and mixed thoroughly. The supernatant was taken and added to the labeled new 96-well plate after centrifugation at 1200 r/min for 5 min. Then, 50 μL of LDH working solution was added to each well and mixed and incubated at room temperature for 30 min. The optical density was measured at OD490 nm with a microplate reader, and the mortality rate of Raji cells was calculated using the following common formula: mortality rate (%) = {[(sample well OD490 nm-blank well OD490 nm) − (sample control well OD490 nm-blank well OD490 nm)]/(maximum enzyme activity control well OD490 nm-blank well OD490 nm)} × 100%.

### 2.10. Detection of the Effect of Anti-CD20/CD3 BsNb on Cytokine Secretion by PBMCs

Raji cells with 5000 cells per well and PBMCs with 50000 cells per well were placed in PerkinElmer Cell Carrier 96-well plates and incubated at 37 °C for 12 h. PBMCs were added to Raji cells at an effector: target ratio of 10:1 as the experimental group. The PBMCs group was added with the same volume of the medium as the control group. Then anti-CD20/CD3 BsNb was added at final concentrations of 0 μg/mL, 5 μg/mL, 10 μg/mL, 20 μg/mL, 40 μg/mL and 80 μg/mL, respectively. Three parallel wells were set in each group and incubated at 37 °C for 24 h. The supernatant of the culture medium was collected by centrifugation, and the level of IFN-γ in the supernatant of the culture medium was detected according to the instructions of the human IFN-γ sandwich ELISA kit. The OD450 nm value was determined with a microplate reader at OD450 nm. The level of IFN-γ was calculated from the standard curve.

### 2.11. Stability of Anti-CD20/CD3 BsNb in Serum

The stability of BsNb was tested by mixing the BsNb with human serum and incubating for different times, then evaluating the changes in its binding activity to CD20 by ELISA. Briefly, 2 μg of anti-CD20/CD3 BsNb in each sample was incubated with human serum from healthy donors and incubated at 37 °C. The incubated samples were harvested at 3 h, 6 h, 12 h, 24 h, and 48 h, and frozen at −20 °C. The samples just added in serum were frozen at −20 °C immediately to represent 0 h and served as the control group. The samples at each time point were used as primary antibodies to add to the 96-well Stripwell™ microplates precoated with 2.5 μg/mL human CD20 extracellular polypeptide. The results were determined by using an HRP-conjugated 6×His-tagged mouse monoclonal antibody (Proteintech Group, Wuhan, China) as a secondary antibody at a concentration of 1:10,000. After adding TMB substrates, the plate was read at OD450 nm in a microplate reader.

### 2.12. Statistical Analysis

GraphPad Prism 6.0 (GraphPad Software, Inc., La Jolla, CA, USA) was used for statistical analysis. Each experiment was independently repeated 3 times, and the measurement data are expressed as the mean ± SD. One-way ANOVA was used for the differences between multiple groups, and Lsd was used for pairwise comparisons between groups. * *p* values < 0.05 were considered to be statistically significant.

## 3. Results

### 3.1. Construction of the CD20-Specific Phage Antibody Library

In the ELISA, CD20 peptide was precoated in a 96-well microtiter plate, preimmune serum was used as a negative control, PBS was used as the blank control, and HRP-labeled rabbit anti-camel antibody was used as the secondary antibody. When the dilution of the final immune serum was 1:32,000, the OD450 nm value was still 2.1-fold higher than that of the negative control, as shown in Figure 1, so the antibody titer in the final immune serum was determined to be 1:32,000.

Nested PCR was used to amplify the VHH gene fragment. The cDNA was used as a template to obtain the gene sequence of the leading signal sequence to the CH2 region in the first round of PCR. After 1% agarose gel electrophoresis, the fragment size was approximately 600 bp, as shown in Appendix A. In the second round of PCR, the recovered product of the first PCR amplification was used as the template to amplify the full length of the VHH gene FR1-FR4, and a fragment size of approximately 400 bp was obtained, as shown in Appendix A.

The antibody library was prepared by the transformation of the VHH-ligated phagemid into *E. coli* TG1 competent cells, and the bacterial solution was diluted at multiple ratios from 10^1^ to 10^10^, coated in 2 × YT-AG solid medium, and cultured overnight at 37 °C to count the colonies the next day. The calculated size of the phage antibody library was 1.2 × 10^8^. Twenty-four clones were randomly selected from a 2 × YT-AG solid medium, and 22 clones among them exhibited a target band of approximately 600 bp by PCR amplification, as shown in Appendix A. The transformation efficiency of the antibody library was 91.7%, as calculated.

### 3.2. Enrichment and Screening of a Human CD20-Specific Single-Domain Antibody

Phage ELISA was used to identify the recombinant phage after three rounds of screening. PBS was used as a blank control, the helper phage M13K07 was used as a negative control, and the OD450 nm value of the negative control was 0.105. Fifty-four clones were detected with OD450 nm values much higher than 0.210, as shown in Figure 2. The results indicated that the recombinant phages produced by 54 of the 92 clones had high antigen-binding activity. The six strains with high binding ability were named 11, 30, 71, 72, 83, and 92 and were further expressed, purified, and identified.

### 3.3. Expression and Purification of a CD20-Specific Single-Domain Antibody

The bacteria transformed with the recombinant vector expressing CD20-specific single-domain antibody were cultured and induced, and the cell lysates containing the recombinant protein were obtained by ultrasonication. The cell lysates containing the recombinant protein were purified by using Ni-NTA Sefinose™ Resin. A concentrated band with a size of approximately 20 kDa appeared in SDS–PAGE analysis, which was consistent with the expected molecular weight, as shown in Appendix A.

### 3.4. Binding Activity and Specificity of the sdAb

The binding activity of the six purified recombinant single-domain antibodies was analyzed by ELISA, as shown in Figure 3A. pET-25b-SBP empty vector-transformed bacterial lysate was used as a negative control with an OD450 nm value of 0.105, PBS was used as a blank control, and an OD450 nm value greater than 0.21 was considered positive. When the purified sdAb was diluted to 5 μg/mL and used as a detection antibody, the OD450 nm values of all six antibodies were greater than 3, indicating that the purified sdAb had strong binding activity to human CD20 extracellular peptide. At the same time, the oligopeptide of human β-amyloid (Aβ_1–42_) was selected as the control to verify the specificity of the six purified recombinant single-domain antibodies, as shown in Appendix A. The OD450 nm values of the six antibodies to Aβ_1-42_ were all less than 0.126, indicating that the purified sdAbs did not bind to other nonspecific oligopeptides. The binding affinity of sdAb was verified by ELISA, as shown in Figure 3B, and we found a progressively decreased signal seen at higher dilutions. To detect the specificity of the purified recombinant single-domain antibody, the human CD20 extracellular polypeptide was separated by SDS–PAGE electrophoresis, and the recombinant single-domain antibody was used as the primary antibody, while the HRP-labeled anti-His tag was used as the secondary antibody and identified by the Western blot method, as shown in Appendix A. There was a target band of 6 kDa, which was consistent with the expected molecular weight, indicating that the recombinant single-domain antibody could specifically bind the CD20 extracellular polypeptide.

### 3.5. Immunofluorescence Assay

After the cultured Raji cells were fixed, the recombinant single-domain antibody was used as the primary antibody, and CoraLite^®^ 488-conjugated 6 × His His-Tag Mouse Monoclonal antibody was used as the secondary antibody for visualization and analysis, as shown in Figure 4. The results showed that the Raji cells expressing CD20 molecules on the surface could be specifically stained by the antibody and produced bright green fluorescence, which formed a ring of fluorescence around the cell periphery (Figure 4A). However, the control group did not show any fluorescence signal in the absence of the single-domain antibody used for detection (Figure 4B) or by fixing the unrelated cell line HEK293A, which did not express CD20 molecule on the surface, and then detected with the single-domain antibody (Figure 4C). These results further proved that sdAbs could recognize and bind to the CD20 molecules on the surface of Raji cells in their native form.

### 3.6. Construction, Expression, and Purification of Bispecific Nanobodies Targeting CD3 and CD20

The synthesized antibody gene was ligated into the pET-22b (+) vector, and the ligated product was transformed into *E. coli* BL21 (DE3) competent cells. Then, the plasmid was extracted and identified by double digestion with *Not* I and *Nco* I, as shown in Appendix A. The recombinant plasmid was digested to obtain a target band of approximately 830 bp and a vector band of approximately 5400 bp, which confirmed the insertion of the antibody gene. The expression of the antibody was induced by IPTG, and recombinant BsNb with a molecular weight of approximately 32 kDa was observed in SDS–PAGE analysis, as shown in Figure 5. The recombinant BsNb was purified by Ni-NTA Sefinose™ Resin.

### 3.7. Identification of the Binding Activity of Anti-CD20/CD3 BsNb to the CD20 Peptide

ELISA was applied to identify the binding activity of purified recombinant BsNb. The microplate was precoated with CD20 extracellular peptide, and gradient-diluted recombinant BsNb was used as the detection antibody, while pET-22b (+) empty vector-transformed bacterial lysates were used as a negative control, as shown in Figure 6. The results showed that the binding of recombinant BsNb to CD20 extracellular peptides occurred in a concentration-dependent manner, and the binding activity became weaker as the concentration of recombinant BsNb decreased.

### 3.8. Flow Cytometry Analysis

The binding activity of anti-CD20/CD3 BsNb to CD20 and CD3 was evaluated by flow cytometry analysis. The CD20-expressing lymphoma cell line Raji cells, CD3-expressing human PBMCs and Raji cells, and PBMCs mixed with the same number of cells were incubated with FITC-labeled anti-CD20/CD3 BsNb. Binding of anti-CD20/CD3 BsNb to Raji cells (Figure 7A), freshly isolated PBMCs (Figure 7B), and a mixed group of Raji cells and PBMCs (Figure 7C) was observed. The results showed that FITC-labeled anti-CD20/CD3 BsNb could bind to Raji cells and freshly isolated PBMCs independently or in the mixed cells.

### 3.9. LDH Release Assay Detects Anti-CD20/CD3 BsNb-Mediated Cytotoxicity of PBMCs In Vitro

The LDH method was used to determine the killing activity of anti-CD20/CD3 BsNb-mediated PBMCs on tumor cells, as shown in Figure 8. Various concentrations of anti-CD20/CD3 BsNb showed a killing effect from 5% to 30% compared with the control group and showed significant concentration dependency. Higher killing effects were observed with the high effector to target ratio at the same antibody concentration, and the best killing effects were shown at the higher concentration of antibody with the same effector to target ratio. All concentrations of anti-CD20/CD3 BsNb significantly promoted the killing effect of PBMCs on Raji cells compared with the control group, with statistically significant differences (*p* < 0.01). The results suggest that the cytotoxic effect of anti-CD20/CD3 BsNb-mediated effector cells is dependent both on the effector to target ratio and anti-CD20/CD3 BsNb concentration.

### 3.10. Detection of the Effect of Anti-CD20/CD3 BsNb on Cytokine Secretion by PBMCs

An ELISA quantitative method was used to detect the cytokine secretion of PBMCs stimulated by anti-CD20/CD3 BsNb when the target cells were incubated with effector cells, and the cytokine secretion of PBMCs stimulated by anti-CD20/CD3 BsNb when effector cells were present, as shown in Figure 9. The level of IFN-γ secreted by PBMCs and stimulated by anti-CD20/CD3 BsNb was increased significantly (*p* < 0.01) in a concentration-dependent manner in the presence of target cells. These results indicated that the activation of T cells into killer effector cells induced by anti-CD20/CD3 BsNb was dose dependent in the presence of tumor cells. Meanwhile, we also observed a dose-dependent increase in IFN-γ secretion in the control group when the anti-CD20/CD3 BsNb was incubated alone with PBMCs, and the level of IFN-γ secretion was approximately 50–60% of the experimental group.

### 3.11. The Stability of Anti-CD20/CD3 BsNb in Serum Was Detected by ELISA

Therapeutic antibodies are obtained through complex biotechnology, and in order to safely use them as therapeutic agents, it is necessary to determine the stability of protein-based drugs in addition to studying their functional activity in vitro. Therefore, we determined the stability of anti-CD20/CD3 BsNb in human serum at 37 °C for different incubation times, as shown in Figure 10. The stability of 3 h, 6 h, 12 h, 24 h, and 48 h was compared with the negative control. The results showed that anti-CD20/CD3 BsNb could still maintain suitable stability after 48 h incubation in human serum at 37 °C, and the binding activity was still retained at 80% if compared with the control group, as shown in Figure 10.

## 4. Discussion

B-cell lymphoma is a heterogeneous disease and is associated with immunodeficiency and other factors. In recent years, the issues of treatment options and the development of new target drugs for these diseases have become hot topics in medical research [12].

Traditional CHOP (combination of cyclophosphamide, hydroxydaunorubicin, oncovin, and prednisone) chemotherapy can inhibit the disease and is used as a common treatment for non-Hodgkin’s lymphoma, but the effect is poor and accompanied by serious side effects. CD20 is the ideal target antigen of B-cell lymphoma, and the combination therapy with a CD20-specific monoclonal antibody, such as rituximab with CHOP, is often used as first-line therapy for both NHL and other non-Hodgkin’s lymphomas clinically, which can effectively improve the activity of T cells and improve the immune function of the body and the therapeutic effect against the disease. Along with the development of antibody engineering, small antibody fragments called single-domain antibodies or nanobodies from Camelidae and cartilaginous fish have persistently received attention in the fields of scientific research and drug discovery. A single-domain antibody contains only the domain of VHH and does not contain the Fc segment of traditional antibodies, so it can avoid the complement reaction caused by the Fc segment. Experimental studies have proven that sdAbs generally do not cause an immune response in the body. This indicates that sdAb has less immunogenicity and is easier to apply in humanization and clinical practice [25]. Due to their small molecular weight and flexibility, single-domain antibodies are currently being used to construct bispecific antibodies or multispecific antibodies.

Bispecific antibodies are antibodies with two different specific antigen-binding sites that can recognize both tumor cell-associated antigens and immune cell surface antigens [26]. One important mechanism of bispecific antibodies in tumor therapy is to mediate immune cell killing by simultaneously targeting tumor antigens with T-cell surface CD3 molecules and activating T cells to directly kill target cells by releasing granzyme and perforin. Activated T cells can also secrete a variety of cytokines to recruit immune helper cells (e.g., NK cells and macrophages), thereby enhancing tumor cell killing effects [27]. BsAbs can be manufactured in several structural formats. Early studies mainly used the IgG-like format, which retains the traditional monoclonal antibody structure of two Fab arms and one Fc region, except that the two Fab sites bind different antigens. However, this format relies on random chance to form usable BsAbs and can be inefficient [28,29]. The later developed BsAbs were mainly of the non-IgG-like format, which consisted only of the Fab regions or single chain variable fragments (scFvs). This format was much smaller in size and relatively simple in design. However, the mismatch between the VH and VL domains and the tendency to aggregate during expression and purification were the main drawbacks of the format [30]. The novel BsAb, which includes CD20/CD3 antibodies, is currently being studied in both aggressive and inactivated non-Hodgkin’s lymphoma, providing specificity for immuno-oncology by enabling the patient’s own T cells to kill malignant B cells through a different approach. Although there are more than 100 academically and industrially derived molecules in clinical development, many of which have similar antigen/receptor targets, these molecules vary greatly in manufacturing and structure. Each form of antibody has its own advantages and disadvantages, but no bispecific nanobody for CD20 and CD3 has been found clinically [31].

Therefore, in this study, a single-domain antibody against CD20, a protein on the surface of human B-cell lymphoma, and a single-domain antibody against CD3, a molecule on the surface of T cells, were fused to construct a novel bispecific single-domain antibody, which confers the advantages of small molecular weight, easy production, and high stability and affinity and can be administered by multiple routes [32].

The anti-CD20/CD3 BsNb was successfully expressed in the prokaryotic expression system, and BsNb was obtained by a simple one-step purification. The results of the ELISA for binding activity showed that the binding of anti-CD20/CD3 BsNb to the CD20 antigen was positively correlated with the antibody concentration to further confirm the specificity and binding activity. A flow cytometric assay was performed. Anti-CD20/CD3 BsNb could simultaneously bind to CD20 molecules expressed on the surface of Raji cells, and CD3 molecules expressed on the surface of T cells in PBMCs.

Studies have shown that CD3 bispecific antibodies can target and bind tumor cells, recruit and activate T cells at tumor sites to specifically kill tumor cells, change the immune microenvironment of tumors, release cytokines such as IL-2, TNF-α, and IFN-γ, and activate other immune cells to exert the body’s own immune function [33,34]. In vitro antitumor experiments showed that anti-CD20/CD3 BsNb could induce PBMCs to play a cytotoxic role by targeting CD20-positive B lymphoma cells in vitro. When the concentration of anti-CD20/CD3 BsNb was 40 μg/mL, and the effector to target ratio was 10:1, the maximum killing rate reached 30.4%. Meanwhile, elevated levels of the secreted cytokine IFN-γ in T cells coincubated with target cells by stimulation with anti-CD20/CD3 BsNb were observed and had suitable stability in human serum at 37 °C. However, a dose-dependent increase in IFN-γ secretion was also observed in the control group when the anti-CD20/CD3 BsNb was incubated with PBMCs, especially in the presence of high concentrations of anti-CD20/CD3 BsNb. Although a certain amount of CD20-expressing B lymphocytes is considered in PBMCs, the increased level of IFN-γ in the presence of high concentrations of the bispecific antibodies remains of concern. Further validation, including in vivo experiments, is needed for the safety and dosimetric relationship of the bispecific antibodies.

## 5. Conclusions

In this study, we successfully constructed a phage antibody library with a size of 1.2 × 10^8^ from a human CD20 polypeptide immunized Bactrian camel. Six strains of phages with high binding ability to human CD20 were named 11, 30, 71, 72, 83, and 92, and induced expression and purification were performed to obtain soluble CD20 single-domain antibodies. The obtained single-domain antibodies could specifically bind to human CD20 polypeptide and cell surface-expressed CD20 molecules in ELISA, Western blot, and cell immunofluorescence assays. 

The anti-CD20/CD3 bispecific nanobody (anti-CD20/CD3 BsNb) was successfully constructed by fusing the anti-CD20 VHH gene with the anti-CD3 VHH gene, and the bispecific single-domain antibody was expressed, purified, and validated. Anti-CD20/CD3 BsNb can specifically bind to CD20 molecules on the surface of human lymphoma Raji cells and CD3 molecules on the surface of T cells in flow cytometry analysis and effectively mediate PBMCs target Raji cells with a killing efficiency of up to 30.4% as measured by the lactate dehydrogenase (LDH) method. The release of hIFN-γ from PBMCs during incubation with anti-CD20/CD3 BsNb was significantly higher than that of the control group (*p* < 0.01). Anti-CD20/CD3 BsNb maintained suitable stability after incubation with human serum at 37 °C for 48 h.

In conclusion, the novel anti-CD20/CD3 BsNb presented in this study, which targets both CD20 and CD3, possesses tumor-killing effects in vitro and has the potential to become a clinical drug candidate. The study provides new evidence to support the potential of single-domain antibody-based bispecific antibodies in antitumor activities. Collectively, our data support further clinical studies with anti-CD20/CD3 BsNb.

## Figures and Tables

**Figure 1 vaccines-10-01335-f001:**
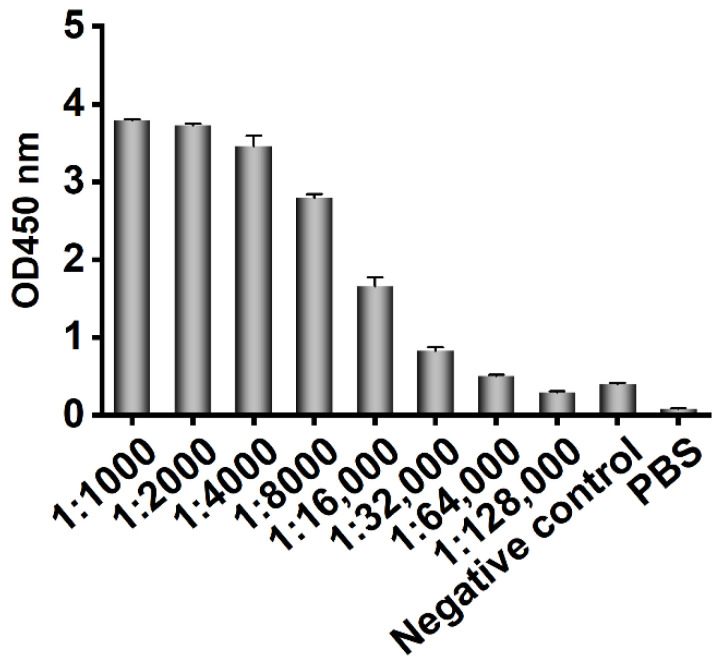
The serum antibody titer of Bactrian camel after immunization was determined by ELISA. CD20 extracellular polypeptide was coated in microplates. Basic serum isolated before immunization was used as negative control, PBS was used as blank control, rabbit anti-camel antibody labeled with HRP was used as a secondary antibody with the working concentration of 1:10,000, and TMB was used for color development. The plate was read at OD450 nm in a microplate reader.

**Figure 2 vaccines-10-01335-f002:**
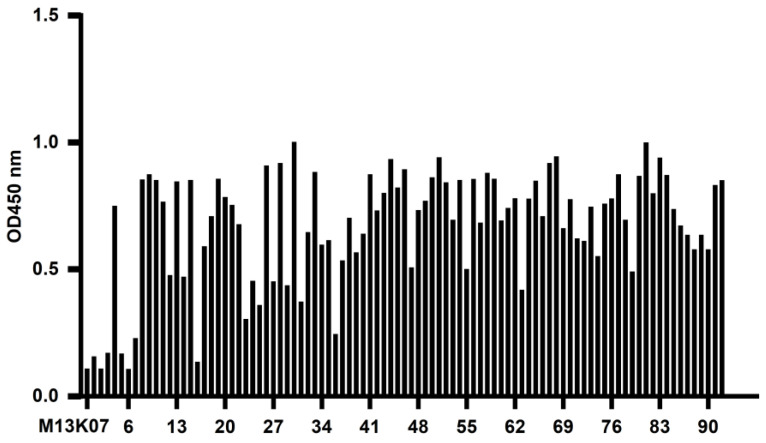
Identification of CD20-specific recombinant phages by phage ELISA. Ninety-two clones of recombinant phage were randomly selected from the third round of panning and added to CD20 extracellular peptide-coated microplates. The bound phage was detected with HRP/anti-M13 monoclonal antibody. PBS was used as a blank control, and helper phage M13K07 was used as a negative control. The OD450 nm of 54 clones detected was 2.1-fold higher than the negative control, indicating that 54 of the 92 recombinant phages had high antigen-binding activity.

**Figure 3 vaccines-10-01335-f003:**
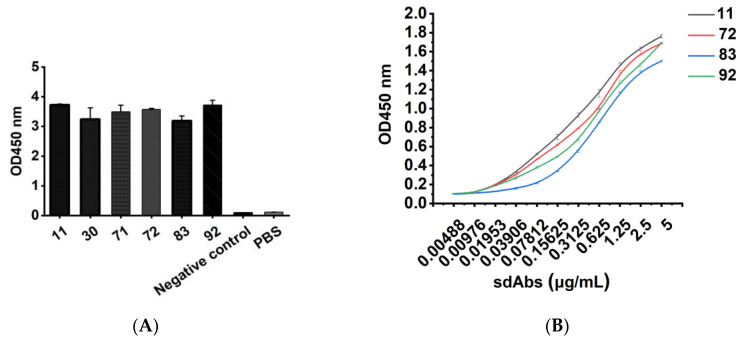
The binding activity of the purified recombinant sdAbs was detected by indirect ELISA. Recombinant sdAbs 11, 30, 31, 71, 72, 83, and 92 were added to CD20 extracellular peptide-coated microplates at a final concentration of 5 μg/mL, and the binding of recombinant sdAb was detected by HRP-conjugated 6 × His-tagged mouse monoclonal antibody. The pET-25b-SBP empty vector-transformed bacterial lysate was used as a negative control, and PBS was used as a blank control (**A**). Recombinant sdAbs 11, 72, 83, and 92 were diluted from 5 μg/mL to 0.00488 μg/mL, and added to microtiter plates precoated with CD20 peptide, and the binding of recombinant sdAbs was detected with HRP-conjugated 6 × His-labeled mouse monoclonal antibody (**B**).

**Figure 4 vaccines-10-01335-f004:**
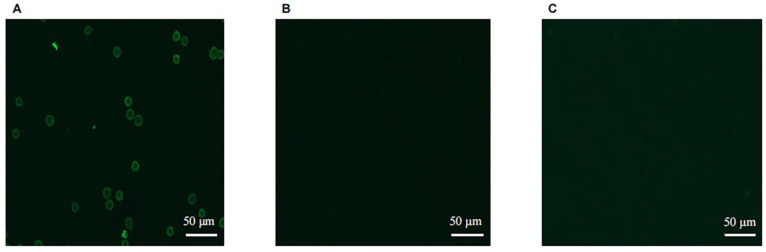
Immunofluorescence analysis of the binding activity of recombinant sdAb to CD20 on the surface of Raji cells. Raji cells were fixed on coverslips, and the recombinant sdAb was used as the primary antibody, and CoraLite^®^ 488-conjugated 6 × His His-Tag Mouse Monoclonal antibody was used as the secondary antibody at a concentration of 1:150. The samples were placed under a confocal microscope (ZEISS, LSM-800) to detect the fluorescence signal. Bright green fluorescence formed a ring of fluorescence around the cell periphery, as shown in (**A**). However, the control group did not show any fluorescence signal in the absence of the single-domain antibody used for detection, as shown in (**B**) or by fixing the unrelated cell line HEK293A, which did not express CD20 molecule on the surface, and then detected with the single-domain antibody (**C**).

**Figure 5 vaccines-10-01335-f005:**
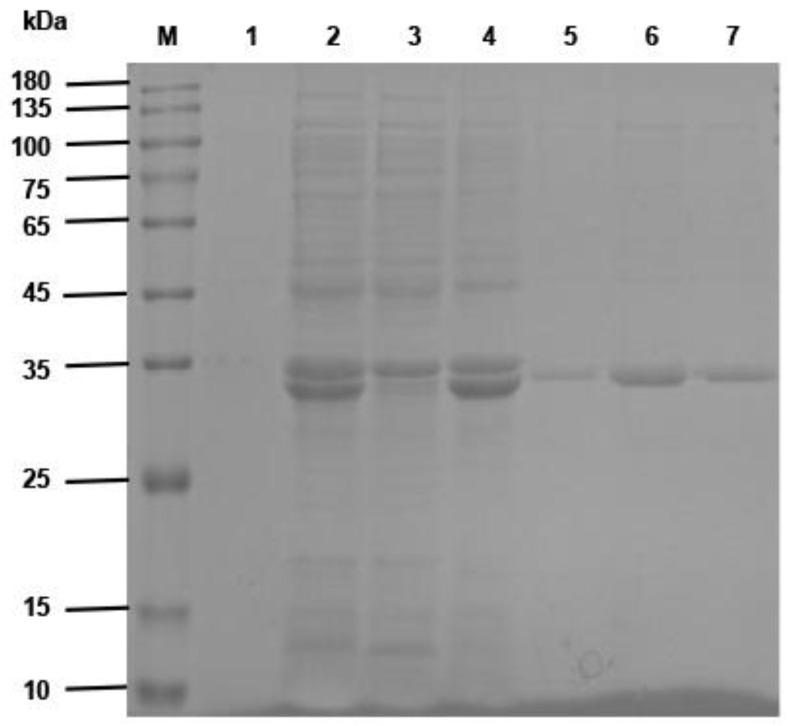
SDS–PAGE analysis of the expression and purification of recombinant BsNb. Recombinant anti-CD20/CD3 BsNb was induced in *E. coli* and purified by Ni-NTA Sefinose™ Resin. A specific protein band of approximately 32 kDa was observed by SDS–PAGE, which is consistent with the expected molecular weight of anti-CD20/CD3 BsNb. Lane M, Protein Marker (10–180 kDa). Lane 1, bacterial lysate before induction. Lane 2, bacterial lysate after induction. Lane 3, the supernatant of bacterial lysate. Lane 4, precipitate of bacterial lysate. Lanes 5–7, elution of anti-CD20/CD3 BsNb from Ni-NTA Sefinose™ Resin.

**Figure 6 vaccines-10-01335-f006:**
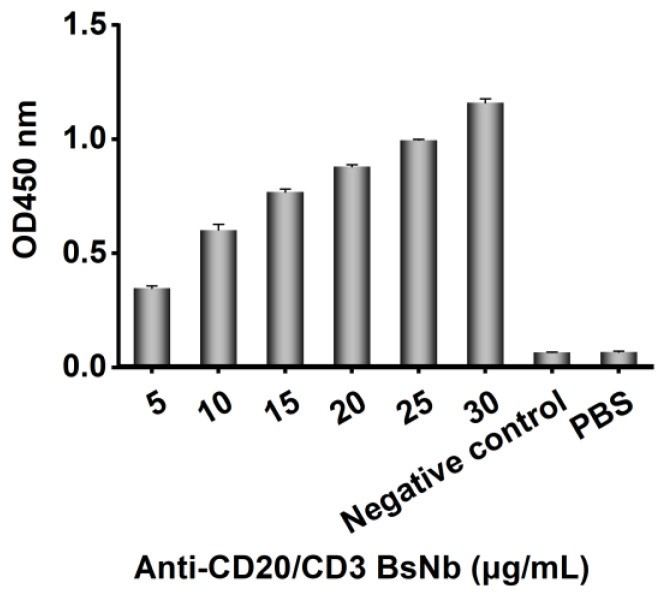
The binding activity of purified recombinant BsNb was measured by ELISA. The binding of recombinant BsNb to the CD20 polypeptide was detected with an HRP-conjugated 6 × His-tagged mouse monoclonal antibody. The pET-22bempty vector-transformed bacterial lysate was used as a negative control.

**Figure 7 vaccines-10-01335-f007:**
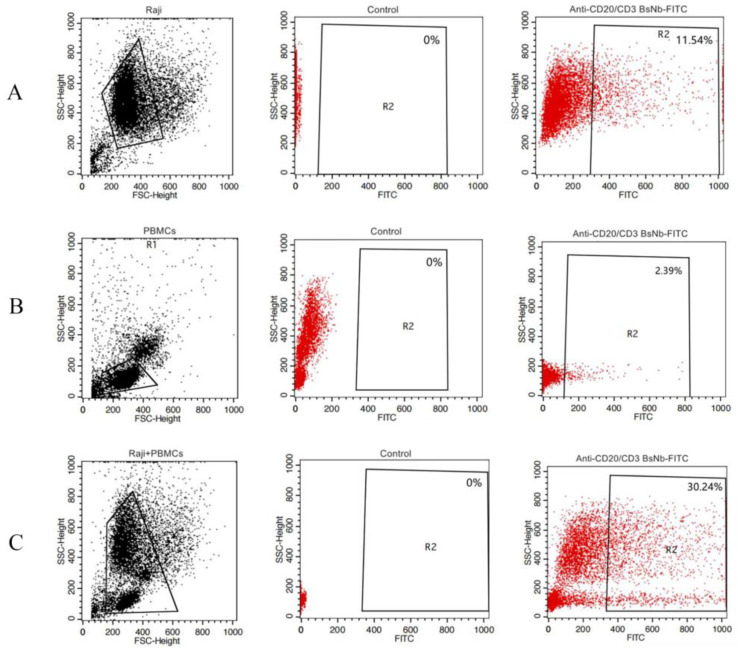
Flow cytometry analysis of the binding activity of anti-CD20/CD3 BsNb. FITC-labeled anti-CD20/CD3 BsNb at a final concentration of 100 μg/mL was added to the CD20-expressing lymphoma cell line Raji, CD3-expressing human PBMCs, as well as Raji and PBMCs, mixed with the same number of cells as the experimental group, and PBS was added to cells as a negative control for the flow cytometry analysis. Binding of anti-CD20/CD3 BsNb to Raji cells (**A**), binding to freshly isolated PBMCs (**B**), and binding to a mixed group of Raji and PBMCs (**C**) were displayed, respectively. The results indicate that the purified anti-CD20/CD3 BsNb antibody could simultaneously bind to Raji cells and PBMCs.

**Figure 8 vaccines-10-01335-f008:**
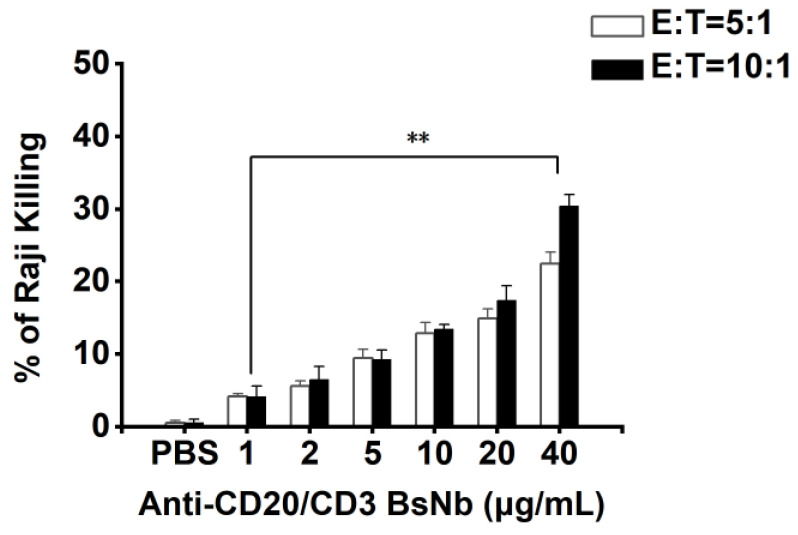
The LDH assay was used to determine the anti-CD20/CD3 BsNb-mediated killing activity of PBMCs against tumor cells. Raji cells were cultured in a PerkinElmer Cell Carrier for 12 h. PBMCs were added to each group in three parallel wells at ratios of effector: target = 5:1 and effector: target = 10:1, and anti-CD20/CD3 BsNb was added to the experimental group to a final concentration of 40 μg/mL, 20 μg/mL, 10 μg/mL, 5 μg/mL, 2 μg/mL, and 1 μg/mL. Then, 100 μL RPMI 1640 medium was added to blank wells, 50 μL of Raji cells + 50 μL of PBMCs were added to sample control wells, and 50 μL of Raji cells + 50 μL of PBMCs + 10 μL of lysate were added to maximum enzyme activity control wells and incubated for 24 h at 37 °C. Then, 10 μL of lysate was added to the target cell maximum release wells 1 h before the end of incubation and mixed thoroughly. The optical density was measured at OD490 nm with a microplate reader, and the mortality rate of Raji cells was calculated using the following common formula: mortality rate (%) = {[(sample well OD490 nm-blank well OD490 nm) − (sample control well OD490 nm-blank well OD490 nm)]/(maximum enzyme activity control well OD490 nm-blank well OD490 nm)} × 100% (** *p* < 0.01).

**Figure 9 vaccines-10-01335-f009:**
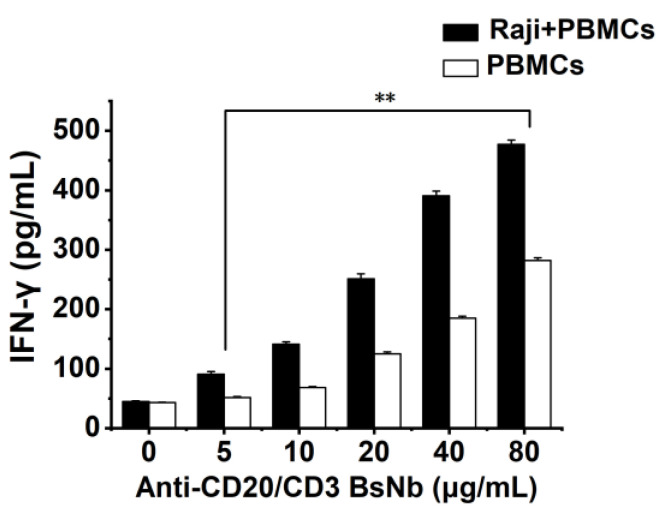
Detection of the effect of anti-CD20/CD3 BsNb on the secretion of cytokines by PBMCs in the presence or absence of Raji cells by ELISA. Raji cells were cultured in PerkinElmer Cell Carrier for 12 h with 3 parallel wells in each group, and effector cell PBMCs with the effector: target ratio = 10:1 were added to the experimental group. PBMCs cells were cultured in PerkinElmer Cell Carrier for 12 h with three parallel wells in each group and the same volume of medium added as the control group. The final concentrations of anti-CD20/CD3 BsNb were 80 μg/mL, 40 μg/mL, 20 μg/mL, 10 μg/mL, 5 μg/mL and 0 μg/mL were added to the Wells. The level of IFN-γ was determined with the ELISA method and calculated from the standard curve (** *p* < 0.01).

**Figure 10 vaccines-10-01335-f010:**
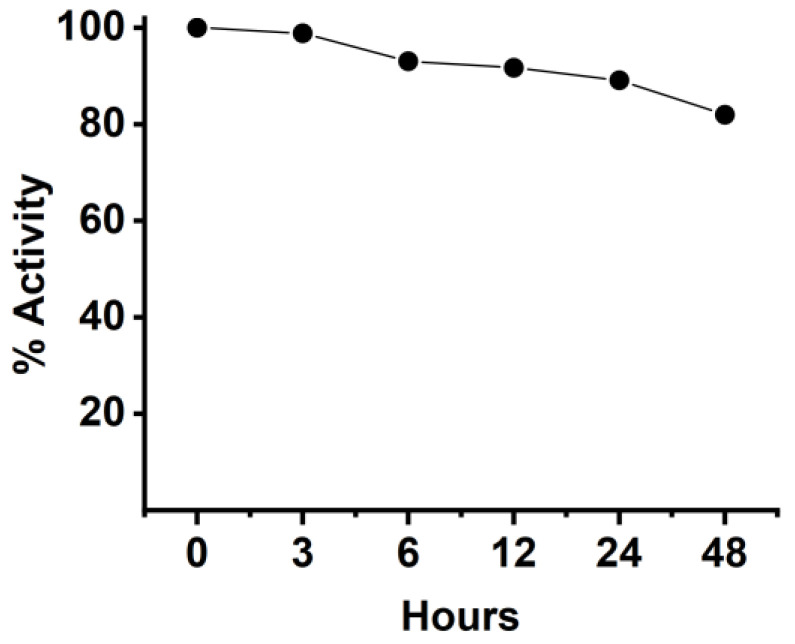
The stability of BsNb was tested by mixing the BsNb with human serum and incubated for different times, then evaluating the changes in its binding activity to CD20 by ELISA. After mixing 2 μg of anti-CD20/CD3 BsNb with human serum, the mixture was determined at different time points after incubation at 37 °C. The mixed BsNbs were used as the primary antibody, and HRP-conjugated 6 × His-tagged mouse monoclonal antibody was used as the secondary antibody to detect the binding activity to CD20 antigen.

**Table 1 vaccines-10-01335-t001:** PCR primers.

Primers	Sequences	Products
P_1_	5′-GGTACGTGCTGTTGAACTGTTCC-3′	600 bp, 900 bp
P_2_	5′-GTCCTGGCTGCTCTTCTACAAGG-3′	
P_3_	5′-AGTTGTTCCTTCTATGCGGCCCAGCCGGCCATGGCTGAKGTBCA GCTGGTGGAGTCTGG-3′	
P_4_	5′-ATTGCGTCAGCTATTAGTGCGGCCGCTGAGGAGACRG TGACCWGGGTCC-3′	400 bp
MP57	5′-TTATGCTTCCGGCTCGTATG-3′	
GIII	5′-CCACAGACAGCCCTCATAG-3′	600 bp

## Data Availability

All data are presented in the manuscript and Appendix A.

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
