# Peer review of "Development of a Bispecific Nanobody Targeting CD20 on B-Cell Lymphoma Cells and CD3 on T Cells"

_vaccines, 2022, doi:10.3390/vaccines10081335_

Round 1
Reviewer 1 Report
The original article " Development of a bispecific nanobody targeting CD20 on B-cell lymphoma cell and CD3 on T cells", presented by Yanlong Liu et al., is a very well presented and methodic description of the generation and characterization of a novel bispecific Nanobody.
The way the authors introduce and explain the different techniques and strategies are very comprehensible.
Even that the field of multispecific antibodies design, characterization and test (in vitro and in vivo) shows different functional models, is always good to see that the technology provides new "weapons" against cancer, lymphoma in this case.
I have few comments and questions and some more minor comments:
- Please, idicate the differences between this manuscript and the recent article published by some of the authors named "Preparation and identification of a single domain antibody specific for adenovirus vectors and its application to the immunoaffinity purification of adenoviruses.", where some of the techniques and results seem to be similar.
- It would be good to see and comment the unspecific activation of the BsNb when incubated alone with PBMCs.
- Structural and functional stability is an important task to check. Incubations at 37ºC with human serum and further Elisa assays and SDS-PAGE gels would indicate the adequate properties of this BsNb. (Please, Check Cuesta AM et al., Improved stability of multivalent antibodies containing the human collagen XV trimerization domain. MAbs. 2012 PMID: 22453098 and Willuda J, et al., High thermal stability is essential for tumor targeting of antibody fragments: engineering of a humanized anti-epithelial glycoprotein-2 (epithelial cell adhesion molecule) single-chain Fv fragment. Cancer Res. 1999 PMID: 10582696).
Minor tasks:
- Line 25, define the acronym BsNb that appear first in line 23; bispecific nanobody, I guess.
- Sentence in lines 23-25 is unclear.
- Define the acronym PBMCs (line 27).
- Define the acronym LDH (line 28).
-Line 120, singular or plural?
- Table 1, PCR primers. Please, clarify to readers the meaning of the non-classical ATCG DNA bases.
- Figures 1, 3, 6, 7, 8 and 9: they should be readable (at least the legends) and their quality must be improved.
- Immunofluorescence images (figure 4) are poor, please, provide better quality and higher zoom images, as well as a scale bar. Add nuclear Dapi staining to detect and localize the cells.
Author Response
The original article " Development of a bispecific nanobody targeting CD20 on B-cell lymphoma cell and CD3 on T cells", presented by Yanlong Liu et al., is a very well presented and methodic description of the generation and characterization of a novel bispecific Nanobody.
The way the authors introduce and explain the different techniques and strategies are very comprehensible.
Even that the field of multispecific antibodies design, characterization and test (in vitro and in vivo) shows different functional models, is always good to see that the technology provides new "weapons" against cancer, lymphoma in this case.
I have few comments and questions and some more minor comments:
- Please, indicate the differences between this manuscript and the recent article published by some of the authors named , where some of the techniques and results seem to be similar.
Author’s response: We sincerely thank the reviewer for their insightful comments and suggestions. First of all, our group recently published a paper titled "Preparation and Identification of a Single Domain Antibody Specific for Adenovirus Vectors and Its Application to the Immunoaffinity of Adenoviruses". Indeed, we applied phage display antibody library technology to screen and prepare antigen-specific nanobody in both articles. This is also a relatively mature nanobody preparation technology established by our research group in the early stage. Although the library construction and screening techniques in the two papers were similar, but in this paper, we constructed bispecific nanobody using the obtained human CD20 nanobody, and systematically identified the in vitro binding activities and tumor cell killing efficacy of bispecific nanobody. The content, techniques and methods used in this paper are much more complicated than the last published paper.
- It would be good to see and comment the unspecific activation of the BsNb when incubated alone with PBMCs.
Author’s response: We thank the reviewer for their insightful comments and suggestions. In the method section “1.8. Flow cytometry analysis”, we incubated anti-CD20/CD3 BsNb alone with PBMCs and confirmed the binding of PBMCs. The results showed in the “3.7. Flow cytometry analysis” and “Figure 7. Flow cytometry analysis of the binding activity of anti-CD20/CD3 BsNb”, and anti-CD20/CD3 BsNb incubated with PBMCs could binding to PBMCs. So when designing the cell killing assay measured by LDH assay, we did not include the anti-CD20/CD3 BsNb incubation with PBMCs, instead just used Raji cells + PBMCs as a control group and ignored the incubation of PBMCs with anti-CD20/CD3 BsNb.
- Structural and functional stability is an important task to check. Incubations at 37ºC with human serum and further Elisa assays and SDS-PAGE gels would indicate the adequate properties of this BsNb. (Please, Check Cuesta AM et al., Improved stability of multivalent antibodies containing the human collagen XV trimerization domain. MAbs. 2012 PMID: 22453098 and Willuda J, et al., High thermal stability is essential for tumor targeting of antibody fragments: engineering of a humanized anti-epithelial glycoprotein-2 (epithelial cell adhesion molecule) single-chain Fv fragment. Cancer Res. 1999 PMID: 10582696).
Author’s response: We would like to thank the reviewers for their suggestions. We agree with the reviewer that structural and functional stability of a new antibody is very important. We referred to paper "Improved Stability of Multivalent antibodies containing the Human Collagen XV Trimerization Domain’’ you suggested, and performed an experiment to identify the stability of anti-CD20/CD3 BsNb and added this section to the manuscript. We added 2 μg of anti-CD20/CD3 BsNb to human serum, and the samples were incubated at 37°C. Samples were harvested at 3 h, 6 h, 12 h, 24 h and 48 h, and frozen at -20°C. The samples just added with serum were frozen at -20°C immediately to represent 0 h and served as the control group. Then, the samples at each time point were used as primary antibody, and HRP-conjugated 6×His-tagged Mouse Monoclonal antibody was used as secondary antibody to detect the binding ability to human CD20 molecules by ELISA. The results showed that the activity of anti-CD20/CD3 BsNb in human serum at 37℃ for 48h was only 17.96% lower than that at 0 h, indicating that it had good stability.
Minor tasks:
- Line 25, define the acronym BsNb that appear first in line 23; bispecific nanobody, I guess.
- Sentence in lines 23-25 is unclear.
- Define the acronym PBMCs (line 27).
- Define the acronym LDH (line 28).
Author’s response: We thank the reviewer for their insightful comments and suggestions. We have redefined the abbreviations of BsNb in line 25, PBMCs in line 27, LDH in line 28 and made changes in the article. We have also made a clearer explanation of the content in line 23-25.
-Line 120, singular or plural?
Author’s response: We appreciate your reminder to change it to plural.
- Table 1, PCR primers. Please, clarify to readers the meaning of the non-classical ATCG DNA bases.
Author’s response: In Table 1, non-classical ATGCs are degenerate bases, where K represents G/T, B represents G/T/C,W represents A/T, and R represents A/G. Due to the annexation of codons, some bases on the designed primers will be marked with degenerate bases, and various bases will be equally distributed during synthesis. The same genes of different organisms are different at the DNA level. It is very difficult to design primers to find a sequence that is 100 percent the same in all organisms. In order for the primer to expand as many sequences as possible, degenerate bases are added.
- Figures 1, 3, 6, 7, 8 and 9: they should be readable (at least the legends) and their quality must be improved.
Author’s response: We thank the reviewer for their insightful comments and suggestions. We readjusted the image and improved the quality of the figures.
- Immunofluorescence images (figure 4) are poor, please, provide better quality and higher zoom images, as well as a scale bar. Add nuclear Dapi staining to detect and localize the cells.
Author’s response: We thank the reviewer for their insightful comments and suggestions. In cellular immunofluorescence, we focused on the binding activity of recombinant sdAb to CD20 on the surface of Raji cells and could detect bright green fluorescence forming a fluorescent ring around the cells. In contrast, the control did not show any fluorescent signal in the absence of the single structural domain antibody used for the assay. Immobilization with the unrelated cell line HEK293A, which does not express CD20 molecules, on the surface followed by detection with a single structural domain antibody also did not show a fluorescent signal. We apologize and regret that we did not have other higher quality plots and did not use DAPI for cell localization because the experiment was performed earlier and was successful. However, we have added the scale bars to the images.
Reviewer 2 Report
I suggest the publication of this manuscript after addressing the following issues:
i) Please improve the quality of figures (letter size etc.);
ii) Please improve the conclusion section adding details and the main results obtained.
Author Response
I suggest the publication of this manuscript after addressing the following issues:
-Please improve the quality of figures (letter size etc.);
Author’s response: We sincerely thank the reviewer for their insightful comments and suggestions. We readjusted the image and improved the quality of the figures.
-Please improve the conclusion section adding details and the main results obtained.
Author’s response: We thank the reviewer for their insightful comments and suggestions. We accept your valuable comments and add details and the results of this experiment in the conclusion of the article.
Round 2
Reviewer 1 Report
Dear Authors,
I thank the answers and corrections done, the manuscript, in my point of view has improved and shows real good charachteristics and potential clinical use.
Nevertheless, I still miss the demostration of the "safety" of the BsNb and the effect on PBMCs. This is a key poiny to be tested if a you are willing for a future clinical use of the BsNb. Hence, I cannot accept the publication of this article unless you provide the data previously required.
Author Response
Author’s response: We sincerely thank the reviewer for their insightful comments and suggestions. We re-detected the secretion level of IFN-γ when anti-CD20/CD3 BsNb was incubated with PBMCs alone in the absence of tumor cells (Raji) as a control. The experimental results have been modified as shown in Figure 9 in the manuscript. The results show that when there is no target cell, anti-CD20 /CD3 BsNb can still activate T cells to release low level of IFN-γ, since there are certain amount of CD20 expressing B lymphocytes in PBMCs. But the level of IFN-γ in control group was much lower than the experimental group in the presence of target cells. We can speculate that the activation of T cells induced by anti-CD20/CD3 BsNb into killer effector cells depends on the presence of tumor cells with a dose-dependent manner, and the anti-CD20/CD3 BsNb is safe when used as a therapeutic agents.
Round 3
Reviewer 1 Report
Dear authors.
Thanks for completing the effect of the BsNb alone.
I must say that the conclusion is very optimistic (lines 515-520) "In contrast, there were only a mild increased level of IFN-γ secretion in the control group that in the absence of target cells. […] These results indicated that the activation of T cells into killer effector cells induced by antiCD20/CD3 BsNb was dose-dependent in the presence of tumor cells.".
The graphic shows a dose-dependent activation, true, but the differences between the presence or the absence of Raji cells does not support such conclusion.
The increase is not mild at all, is about 6 times higher than 0 µg/ml and about 30% lower than the less than 80 µg/ml when compared with Raji+PBMCs.
Therefore, I ask the author to re-write the text accordingly to the obtained data.
Author Response
Author’s response: We sincerely thank the reviewer for their insightful comments and suggestions. We reedited the results in the section “3.9. Detection of the effect of anti-CD20/CD3 BsNb on cytokine secretion by PBMC”, and added a part of content in the end of the discussion section. Please find them in the revised manuscript.